# An optimal dichotic-listening paradigm for the assessment of hemispheric dominance for speech processing

René Westerhausen[1,2]*, Fredrik Samuelsen[2]

**1** Department of Psychology, University of Oslo, Oslo, Norway, **2** Department of Biological and Medical Psychology, University of Bergen, Bergen, Norway

* rene.westerhausen@psykologi.uio.no

**Data Availability Statement:** All 1 datafiles are available from the OSF database (OptDL project: DOI 10.17605/OSF.IO/AJ26N).

## Abstract

Dichotic-listening paradigms are widely accepted as non-invasive tests of hemispheric dominance for language processing and represent a standard diagnostic tool for the assessment of developmental auditory and language disorders. Despite its popularity in research and clinical settings, dichotic paradigms show comparatively low reliability, significantly threatening the validity of conclusions drawn from the results. Thus, the aim of the present work was to design and evaluate a novel, highly reliable dichotic-listening paradigm for the assessment of hemispheric differences. Based on an extensive literature review, the paradigm was optimized to account for the main experimental variables which are known to systematically bias task performance or affect random error variance. The main design principle was to minimize the relevance of higher cognitive functions on task performance in order to obtain stimulus-driven laterality estimates. To this end, the key design features of the paradigm were the use of stop-consonant vowel (CV) syllables as stimulus material, a single stimulus pair per trial presentation mode, and a free recall (single) response instruction. Evaluating a verbal and manual response-format version of the paradigm in a sample of N = 50 healthy participants, we yielded test-retest intra-class correlations of $r_{ICC}$ = .91 and .93 for the two response format versions. These excellent reliability estimates suggest that the optimal paradigm may offer an effective and efficient alternative to currently used paradigms both in research and diagnostic.

## Introduction

Verbal dichotic-listening paradigms offer well-established behavioral tests for the assessment of latent hemispheric differences for language processing [1, 2] and are integral part of test procedures for the diagnosis of auditory processing disorders [3–5]. A significant advantage of dichotic compared with alternative paradigms (e.g., visual-half field techniques or neuroimaging approaches) is the simplicity of the testing procedure which can be easily understood and performed also by young children [e.g., 6, 7], elderly individuals [e.g., 8, 9], or patients with cognitive disabilities [e.g., 10, 11]. That is, in its basic form, pairs of verbal stimuli (e.g., words

**Funding:** The "Department of Psychology, University of Oslo" has supported the work with funds.

**Competing interests:** The authors have declared that no competing interests exist.

or syllables) are presented via headphones, with one of the stimuli presented to the left ear and the other one, simultaneously, to the right ear [12]. Instructed to report the one of the two stimuli which was heard best, participants typically report the right-ear stimulus more frequently, more accurately, and more rapidly than the left-ear stimulus. This right-ear advantage is widely accepted to indicate left hemispheric dominance for speech processing [2, 13, 14] and differences in the magnitude of this right-ear preferences have been related to interhemispheric auditory integration [15–17].

However, many different versions of dichotic-listening paradigms have been suggested [12, 18] and the test-retest reliability of most dichotic paradigms–even of those used for diagnostic purposes–are far from optimal [19, 20]. This shortcoming severely threatens the inferences that can be made using dichotic-listening measures, as the reliability of a test also sets the upper limit of its validity [21]. At the same time, however, substantial differences in reliability estimates between paradigms [19] suggest that certain design features systematically affect the consistency of an individual's dichotic listening task performance. In fact, since the early conceptualization of dichotic listening more than six decades ago [22, 23], a plethora of studies has accumulated a significant amount of evidence on how features of the experimental set-up (e.g., stimulus order, stimulus material) and task instructions lead to systematic response biases [for review see ref. 18]. For example, it has been demonstrated that paradigms instructing participants to selectively attend to only one ear at a time are more difficult for participants to perform than paradigms allowing for a free selection [24]. Also, presenting multiple stimulus pairs per trial increases the working-memory load of the task by requiring the participant to keep the representation of multiple stimuli activated simultaneously, leading to a decreased right-ear advantage [25].

Ignoring these design variables or leaving them uncontrolled introduces error variance to the obtained measures, affecting both the reliability of the obtained laterality measures and the efficiency of the paradigm. In turn, however, considering these variables when designing a dichotic-listening task makes it possible to tailor a paradigm which is optimized for a given purpose. In the present paper, the intention is to design a dichotic listening paradigm optimized for the assessment of hemispheric dominance for speech and language processing. As theoretical framework, we assume that dichotic-listening performance can be best explained using a two-stage model [26–28]: an initial stage leading to a perceptual representation of the two competing stimuli in verbal short-term memory [29, 30], and a second stage characterized by cognitive-control processes which may modulate the initial representation in line with task requirements, resulting in a response selection [31, 32]. It is further assumed that a "true" underlying right-ear preference exists as consequence of left-hemispheric specialization for speech and language processing [33, 34]. That is, in the initial stage, the right-ear stimulus can be expected to be more salient than the left-ear stimulus. An optimal paradigm to assess this underlying perceptual, "built-in" laterality, consequently needs to assure both an unbiased initial stimulus representation and a cognitively unaltered response selection during second stage processing [27]. Based on recent literature review [18], we here aim to create and evaluate such an optimal paradigm by following the eight design features listed in Table 1.

However, while many design features of dichotic paradigms have been studied systematically, it appears somewhat surprising that little is known about the effect of the response format on task performance [18]. When using a verbal-response format, the participant is typically instructed to repeat orally the stimulus she/he perceives after each trial, and the response is either ad-hoc scored by an experimenter or recorded for later decoding [25, 35]. Manual responses are typically collected via button press on a keyboard, special response box, or computer mouse [28, 36]. Both alternatives come with theoretical advantages and disadvantages [18]. For example, the verbal-response format carries the risk of increasing error variance

**Table 1. Design features of the optimal dichotic-listening paradigm and arguments for their implementation.**

| # | Design feature | Argumentation |
|---|---|---|
| 1 | Stop consonant-vowel (CV) syllables as stimulus material | Proven to be valid test material [38]. CV show higher reliability than numeric or non-numeric words as stimulus material [19] |
| 2 | Pair only CV stimuli from the same voicing category | Same-voicing category pairs are more likely to fuse into one percept than mixed pairs; reduces relevance of attention and cognitive-control processes [28] |
| 3 | Includes binaural (diotic) trials | Allows to demonstrate stimulus appropriateness, i.e. whether participant is able to identify the used stimulus material [18] |
| 4 | Alternating trials of voiced and unvoiced stimulus pairs | Limits negative-priming effects [39] by preventing stimulus repetition between consecutive trials |
| 5 | Paradigm length of 120 dichotic trials | Previous studies indicate reliability estimates >.80 mostly for paradigms using 120 trials [18] |
| 6 | Single-stimulus pair per trial; single, immediate response | Minimizes working-memory load compared to multi-stimulus trials [25] and delayed response paradigms [36] |
| 7 | Free-recall instruction | Reduces task difficulty and relevance of cognitive-control processes compared to selective attention instructions [24] |
| 8 | All stimulus pairs are presented in both orientations (i.e., left-right ear and right-left orientation) in identical frequency | Averages otherwise uncontrolled biases across these trials (e.g., item difficulty effects; see [40, 41]) |

(a) for a detailed discussion refer to [18]

in the recorded data as the response has to be decoded by the experimenter. As compared to a verbal version, using a manual response format changes the cognitive demands of the paradigm as it demands additional response mapping and selection processes including visual-motor coordination [37]. However, an empirical comparison of the two response formats is missing from the literature and it is to date unknown if the reliability of a dichotic paradigm is affected by the response format utilized.

In summary, the aim of the present study was to develop and test a new dichotic-listening paradigm for the assessment of hemispheric dominance which considers the main design feature known to affect task performance and yields high test-retest reliability. To evaluate the retest reliability of this novel paradigm, participants had to complete the full paradigm four times: each two times using verbal and manual response format. This setup also allows evaluating possible effects of the response format on laterality estimates.

## Material and methods

### Participants

The final sample consisted of N = 50 participants (n = 30 female, 60%; age: mean ± s.d. was 25.0 ±4.2 years, range 19 to 39 years) recruited among students and employees of the Universities of Oslo (n = 17) and Bergen (n = 33), Norway. All participants were right-handed (as verified in Edinburgh Handedness Inventory, [42]) and had no history of neurological or psychiatric conditions (as verified in self-report). The participants were fluent speakers of Norwegian (42 native speakers, 8 non-native speakers) and high identification rate (mean: 98.3 ± 2.2%) of the used stimuli when presented diotically (see next section) assured that the used stimulus material was appropriate for all included participants. Hearing acuity was assessed in an audiometric screening (Oscilla USB-300, Inmedico, Lystrup, Denmark) for pure tones of 250, 500, 1000, 2000, and 3000 Hz. All included participants had an absolute interaural

threshold difference smaller than 10 dB (range from -8 to 8 dB in favor of the right and left ear, respectively; mean = -0.6 ± 4.1) and absolute threshold of smaller than 25 dB (mean left ear: 4.7 ± 7.5 dB; right ear: 5.2 ± 7.2 dB) across the tested frequencies. For matter of completeness, the final sample excluded two participants: one who identified only 59 of the maximum 72 diotic stimuli correctly (81.9%), and one participant with an interaural acuity difference of 16 dB.

The study was approved by the ethical review board of the Department of Psychology, University of Oslo, and written informed consent was obtained from all participants.

## Stimulus material and paradigm design

Six stop-consonant-vowel (CV) syllables served as stimulus material; three of voiced (/ba/, /da/, /ga/) and three of unvoiced articulation (/pa/, /ta/, /ka/). The syllables were natural recordings of a male Norwegian voice actor spoken in constant intonation and intensity. CV-syllables were preferred over alternatives (e.g., rhyming words) as they tend to show higher reliability than other stimulus material [19] and the validity of CV-based test has been repeatedly confirmed (e.g, using the Wada test [38], functional MRI [43], and in studies on surgical patients [44]; see point 1, Table 1). Furthermore, to increase the likelihood of perceptual fusion, syllables of the same voicing category were paired exclusively ([28], point 2, Table 1). That is, the dichotic stimulus pairs consisted of either the six possible pairwise combinations of each the three voiced syllables (e.g., /ba/-/da/, /da/-/ba/) or the three unvoiced syllables (e.g., /pa/-/ta/, /ta/-/pa/). The six diotic pairs with the same stimulus presented to both ears (e.g., /ba/-/ba/, /pa/-/pa/, etc.) were also included as control trials (see point 3, Table 1).

The paradigm started with a test block of 10 trials to familiarize the participants with the setup and response procedure. This was followed by three experimental blocks, each containing 46 stimulus pairs (40 dichotic and 6 diotic). Thus, one run of the experiment consists of a total of 120 dichotic (i.e., ten complete presentations of the full set of twelve stimulus pairs) and 18 diotic pairs (i.e., three times the full set of six). The order of trials within each block was pseudorandomized so that consecutive trials did not contain syllables from the same voicing category, preventing direct negative-priming effects ([39], point 4, Table1) and the order was the same for each participant. All stimulus pairs were presented in both orientations (i.e., left-right ear and right-left orientation) in identical frequency in one run of the paradigm (point 8, Table 1). The number of 120 dichotic trials was selected as a literature review has suggested that high reliability can be expected for 90 to 120 trials ([18], see point 5, Table 1).

Each trial was 4000 ms long. A preparation interval of 1000 ms was followed by the stimulus presentation (500 ms) and a response-collection interval of 2500 ms. The stimulus-onset asynchrony was fixed to 4000 ms. A fixation cross (+) was presented in the center of the PC monitor at trial onset but was briefly replaced by a circle (o) to confirm that a response was registered. One trial consisted of a single dichotic stimulus presentation and participants were instructed to report immediately after each stimulus presentation (point 6, Table 1) the one stimulus heard best (free-recall instruction; point 7, Table 1). Response collection was performed via a keyboard using the number pad, on which six response keys (numbers 1 to 6) were marked with the syllable names. There was no possibility to correct a response once given.

The paradigm was administered using both a verbal and a manual response format. The manual response required the participants to log the response themselves. In the verbal version, the participants repeated the best heard syllable aloud after each trial and the experimenter recorded the response. In order not to bias the experimenter's perception of the participant's response, the experimenter was blind to which syllables were presented on each trial. One run of the paradigm took approximately 12 min including instructions, and short breaks between the blocks. The paradigm is available as part of the OptDL Project on the OSF platform [45].

For further analyses, the relative difference between the number of correctly recalled left- ($L_c$) and right-ear ($R_c$) stimuli was determined as laterality index (i.e., LI $= (R_c−L_c)/(R_c+L_c)$) per person and test run.

## Procedure

Each participant completed the paradigm four times, each twice with verbal and manual response format, respectively. The order followed an AABB design, whereby the order was balanced (odd-even by order of recruitment) between individuals, so that n = 26 started with the verbal and n = 24 with the manual version. The setup and test procedure were parallelized at the two sites of data collection (Bergen and Oslo). The same audiometer system and headphone models (Sennheiser HD280) were used at both sites and testing took place in a quiet test room. However, differences in the exact equipment (i.e., computer build, keyboard) were unavoidable. While at both sites E-Prime (Psychology Software Tools, Sharpsburg, PA) was used to control the experiment, version 2.0 was available in Bergen and version 3.0 in Oslo. Nevertheless, we have no reason to believe that these equipment differences had a relevant effect on the present results. Firstly, the effects of interest in the present study are based on comparisons within participants. Secondly, exact timing was not critical as only accuracy and not reaction time data was used.

## Statistical analysis

Test-retest reliability was assessed separately for the LI, $L_c$, and $R_c$ as obtained from the verbal- and the manual-response format paradigm by using intra-class correlations ($r_{ICC}$). A two-way mixed-models aiming for absolute agreement and estimating the coefficient for a single measure (ICC(A,1) model according to [46]) was employed. That is, a two-way model was chosen as test-retest effects were expected (e.g., due to familiarization with the test procedure) so that the second measure might differ systematically from the first. The observations/participants were randomly assigned. while the measurements were determined by the research question (fixed), constituting a mixed design. As test length is a major determinant of reliability [21], we calculated reliability for the full length paradigm of 120 trials, as well as for the first 40 (i.e., blocks 1 vs. 2) and 80 trials (i.e., block 1 and 2 vs. block 3 and 4) to evaluate the test length effect for the present paradigm. Comparability of verbal and manual response format was evaluated (a) by intra-class correlations between the runs of the two versions, and (b) using a mixed-model analysis of variance (ANOVA). The ANOVA was a three-factorial design, including the repeated-measure factors Repetition (first vs. second run of the paradigm) and Response Format (verbal vs. manual), as well as the between participant factor Sex. The dependent variable was the LI. Statistical analyses were conducted in IBM SPSS Statistics (version 25.0, IBM Corp. Armonk, NY) and GPower (www.gpower.hhu.de, version 3.1) was used for test power calculation. Effect sizes were expressed as Cohen's $d$ or proportion explained variance ($\eta^2$). The data can be downloaded from the associated OSF platform [45].

## Results

Mean laterality for the verbal response format was LI = 33.2 (± 22.9) at the first run of the paradigm and LI = 37.4 (± 22.3) upon retest. For the manual paradigm the mean was LI = 29.9 (± 22.7) at first testing and LI = 34.4 (± 22.8) for the retest. All four LI were significantly larger than 0 (all $t(49) > 9.3$; all $p < .001$, all Cohen's $d > 1.27$). The mean differences between the two test runs were accordingly 4.4 (± 7.3) for the verbal response paradigm and 4.2 (±8.6) for the manual (both $t(49) > 3.5$, all $p < .001$, all $d > 0.5$). Mean values for the correct report of left- and right-ear stimuli are presented in Table 2.

**Table 2. Mean values, standard deviation (s.d.), and reliability ($r_{ICC}$) [a] of correct recall of left- ($L_c$) and right-ear stimuli ($R_c$).**

| Response format | Run | Left ear | | | Right ear | | |
|---|---|---|---|---|---|---|---|
| | | mean | s.d. | $r_{ICC}$ (CI$_{95\%}$) | mean | sd | $r_{ICC}$ (CI$_{95\%}$) |
| Manual | First | 40.9 | 13.5 | .93 (.82;97) | 75.9 | 13.7 | .93 (.82; 97) |
| | Retest | 38.3 | 13.3 | | 78.6 | 13.9 | |
| Verbal | First | 39.1 | 13.5 | .92 (.85;.96) | 78.1 | 13.9 | .88 (.73; .94) |
| | Retest | 37.0 | 13.2 | | 81.4 | 13.4 | |

(a) Intra-class correlations determined as ICC(A,1)

The analysis of the full paradigm (i.e. 120 trials) yielded a reliability of $r_{ICC}$ = .93 (95% confidence interval, CI$_{95\%}$: .82 to .97) and .91 (CI$_{95\%}$: .82 to .96) for the verbal and manual response paradigm, respectively (all $p$ < .001). Fig 1 shows the scatterplots illustrating the correspondence between the first and the second measure of both paradigm versions. Considering a single block of 40 trials of the verbal-response version, the reliability was $r_{ICC}$ = .65 (CI$_{95\%}$: .08 to .85), considering two blocks (i.e., 80 trials) it was $r_{ICC}$ = .86 (CI$_{95\%}$: .76 to .92). For a single block in the manual paradigm the reliability was $r_{ICC}$ = .64 (CI$_{95\%}$: .25 to .82); for two blocks it was $r_{ICC}$ = .90 (CI$_{95\%}$: .82 to .94). Reliability estimates for correct report of left- and right-ear stimuli are provided in Table 2.

Comparing the verbal- and manual-response version (120 trial paradigm), we found a $r_{ICC}$ = .71 (CI$_{95\%}$: .54 to .82) of the LI of the first verbal test with the LI of the first manual test, and of $r_{ICC}$ = .77 (CI$_{95\%}$: .63 to .86) with the second manual test. For the second run of the verbal test the correlations were $r_{ICC}$ = .75 (CI$_{95\%}$: .53 to .86) with the first, and $r_{ICC}$ = .83 (CI$_{95\%}$: .72 to .90) with the second manual test run.

A significant main effect of Repetition ($F(1,48)$ = 30.7, $p$ < .001) was found in the ANOVA, although with a small effect size ($\eta^2$ = 0.01), indicative for a stronger LI at retest compared to first testing (across both response formats). The main effect of Response Format was not significant ($F(1,48)$ = 2.28, $p$ = .14, $\eta^2$<0.01). A sensitivity power analysis (at 5% alpha probability, and with $r$ = .80 correlation of the repeated measures) suggests that medium to large effects (>3% variance explained) can be excluded with a power of .80. Neither the interaction of Repetition and Response Format ($F(1,48)$ = 0.04, $p$ = 0.84, $\eta^2$ < 0.01) nor any of the main or interaction effects including the factor Sex (all $F(1,48)$ <1, all $p$>.50, all $\eta^2$ <0.01) were significant.

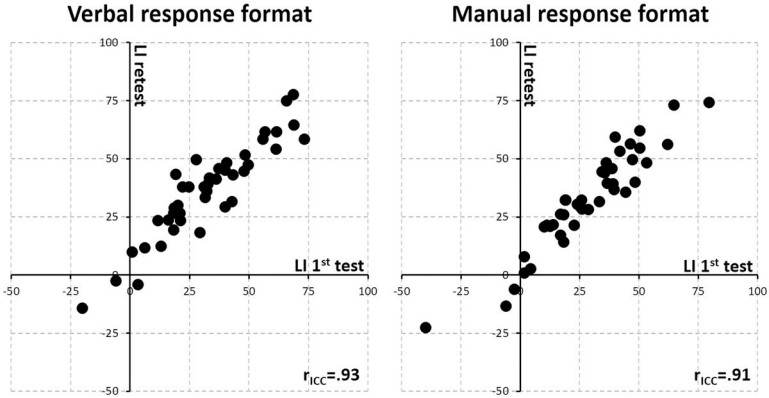

**Fig 1. Scatterplots showing the test-retest correlation for the verbal- and manual-response format version of the optimal paradigm (full length, i.e. 120 dichotic trials).** $r_{icc}$ = intra-class correlation, using mixed model, considering absolute agreement, and a single measure.

## Discussion

Irrespective of response format the here introduced paradigm shows excellent retest reliability. With values of .91 and .93, the full-length paradigm of 120 trials yielded reliability estimates which are in the upper end of what is typically reported for dichotic-listening paradigms [19]. That is, in the past, test-retest correlations between a minimum of $r = .63$ ([47], using rhyming words as stimuli) and a maximum of $r = .91$ ([48]; using vowel-consonant-vowel stimuli) have been reported for paradigms of the same length. Even as a shortened version of 80 trials, the present paradigm reaches reliability estimates of .86 and .90, which can be considered "good" to "excellent" [49]. In fact, previous studies (even testing with 90 or 100 trials) commonly yield retest correlations below the lower 95%-confidence bound for the reliability estimate of the present 80-trial paradigm [cf. 19] suggesting that the difference is significant. For example, Speaks et al. [50]–using CV syllables as stimuli but demanding to report both stimuli presented per trial–report a retest correlation of $r = .71$ in a 90 trial paradigm. However, the confidence intervals for the 80-trials version are wider than for the full length paradigm, including reliabilities below .80 suggesting that the full length version should be preferred. Nevertheless, the above comparison of our results to those of various previous paradigms indicates superior reliability of the present paradigm irrespective of test length. Thus, it appears the design features implemented here indeed improve reliability estimates for LIs measured with dichotic listening.

According to the Spearman-Brown formula for test length adjustment, the number of trials is a major determinant of reliability [21] and, accordingly, we here found an increase in reliability from 40, via 80, to 120 trials. Thus, one might speculate that an additional increase in test length would yield further reliability improvement. However, the relationship between test length and reliability is not linear but rather shows diminishing returns with increasing reliability. Applying the Spearman-Brown formula in the present case, an increase to 160 or 200 trials can be predicted to only marginally increase the reliability. For example, considering the verbal version of the paradigm, reliability would be predicted to increase from .93 to .95 and .96, respectively. Furthermore, previous studies using paradigms longer than 120 trials do not report a substantial improvement or even see a reduction in the reliability estimates for higher trial numbers [50, 51]. Using an excessive number of trials also carries the risk of tiring the participants with adverse effects for test performance and reliability, especially in patient samples. Taken together, the here evaluated 120-trial paradigm seems to combine acceptable test length and excellent reliability estimates.

For matter of completeness, the above evaluation of the present paradigm in the context of previous paradigms, also has to consider that the reliability estimates of previous studies are usually reported as product-moment correlations. Product-moment correlations have two shortcomings for estimating reliability [46]. Firstly, they do not account for the fact that the correlated measures represent the same "class" (i.e., the measures are repeated measures of the same variable collected with the same test). Secondly, product-moment correlations ignore mean differences between the two measures as the calculation of the correlation coefficient only considers the covariance of the variables. Addressing both issues, the here reported intra-class correlation for absolute agreement represent more appropriate but also slightly more conservative estimates of retest reliability than previous studies [49]. For illustration, regarding the manual paradigm, the product-moment correlation would be $r = .95$ as opposed to the $r_{ICC} = .91$ in the present study. Thus, when comparing the present with previously suggested paradigms, it has to be kept in mind that previous studies likely overestimate the "true" reliability of their paradigms.

The intra-class correlations between the measures of the two response format versions were, with values between .71 to .83 well below the reliability estimates. This might suggest

that both paradigms measure slightly different aspects of hemispheric specialization. For example, regarding verbal responses, speech production as part of the response phase might modulate the perceptual laterality, whereas a manual response might not [18]. The oral response also has to be understood and registered by the experimenter, which might be associated with coding errors. Or, using the manual response format, the participant–in addition to identifying the syllables–also has to become familiarized with the response procedure and the response key setup, potentially making the task more difficult. At the same time, the main effect of Response Format was non-significant and small, while the test power was sufficient to exclude substantial mean LI differences between the paradigm versions with some confidence. Nevertheless, the reduced inter-correlations indicate that small differences between verbal and manual response format exist, and it is currently not possible to determine whether one of the two versions is better suited for measuring laterality.

As a limitation, it has to be acknowledge that the here presented reliability estimates are obtained on a sample of adult university students and employees with good hearing acuity and diotic identification rates. Thus, it remains to be established if individuals outside these inclusion criteria, especially clinical or aging populations, also provide reliable data, or whether adjustments to the testing procedure are required. For example, a recent study on an aging sample, rather than excluding individuals with hearing deficits, successfully adjusted the sound level of stimulus presentation to compensate for individual differences in the hearing threshold [9]. Future studies are also required to confirm validity of the paradigm. While past studies strongly suggest good validity of paradigms using CV stimulus material [43, 44] a formal test against more direct indicators of hemispheric dominance remains to be conducted.

In summary, we have designed a highly reliable dichotic-listening paradigm by (a) minimizing the relevance of higher cognitive functions on task performance and (b) optimizing crucial features of the stimulus presentation. Point (a) additionally has the beneficial side effect that individual and group differences in higher cognitive functions are less likely to affect the obtained laterality measures, allowing for a fair testing of clinical or developmental samples. Point (b) optimizes the number of trials required to balance the design, reducing the overall number of trials necessary to yield reliable measures (e.g., by reducing trials which are biased due to negative priming etc.), allowing for economic assessment of laterality, at least in experimental settings.

## Acknowledgments

No conflict of interest exists for the authors. The authors like to thank Kristin Audunsdottir, Nelin Gökbel and Sarjo Kuyateh for supporting the data collection.

## Author Contributions

**Conceptualization:** René Westerhausen.

**Data curation:** René Westerhausen, Fredrik Samuelsen.

**Formal analysis:** René Westerhausen, Fredrik Samuelsen.

**Investigation:** Fredrik Samuelsen.

**Writing – original draft:** René Westerhausen, Fredrik Samuelsen.

**Writing – review & editing:** René Westerhausen.

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
