## [Decision Letter · Decision Letter 0]

22 Apr 2020

PONE-D-20-06948

An optimal dichotic-listening paradigm for the assessment of hemispheric dominance for speech processing

PLOS ONE

Dear Dr Westerhausen,

Thank you for submitting your manuscript to PLOS ONE. After careful consideration, we feel that it has merit but does not fully meet PLOS ONE’s publication criteria as it currently stands. Therefore, we invite you to submit a revised version of the manuscript that addresses the points raised during the review process.

We would appreciate receiving your revised manuscript by Jun 06 2020 11:59PM. To enhance the reproducibility of your results, we recommend that if applicable you deposit your laboratory protocols in protocols.io, where a protocol can be assigned its own identifier (DOI) such that it can be cited independently in the future. For instructions see: http://journals.plos.org/plosone/s/submission-guidelines#loc-laboratory-protocols

We look forward to receiving your revised manuscript.

Kind regards,

Simone Sulpizio

Academic Editor

PLOS ONE

"Funding was provided by the Department of Psychology, University of Oslo, Norway."

Reviewers' comments:

Reviewer's Responses to Questions

**Comments to the Author**

1. Is the manuscript technically sound, and do the data support the conclusions?

Reviewer #1: Partly

Reviewer #2: Yes

2. Has the statistical analysis been performed appropriately and rigorously? 

Reviewer #1: No

Reviewer #2: Yes

3. Have the authors made all data underlying the findings in their manuscript fully available?

Reviewer #1: Yes

Reviewer #2: Yes

4. Is the manuscript presented in an intelligible fashion and written in standard English?

Reviewer #1: Yes

Reviewer #2: Yes

5. Review Comments to the Author

Reviewer #1: Thank you for the opportunity to review this excellent manuscript. The authors have thoughtfully designed an evidence based dichotic listening test that seeks to minimize maximize reliability. Their evidence informed design is well researched and clearly supported by the text. The evaluation is an excellent research design and a significant contribution to the literature. I offer suggestions that I hope will be valuable.

Major issues:

1. The authors should include information about listeners’ absolute scores (right ear and left ear) in addition to laterality index. Because LI uses total correct responses (right plus left) as the denominator, for reliability purposes the number of test items is effectively reduced to whatever number the listener got correct [LI can then be modeled as a binomial]. Knowing the total correct scores are important for readers to be able to draw inferences about generalizability. (i.e., the laterality estimate may be reliable among people with high total scores but have much worse reliability among people with more errors, such as a clinical population.)

2. I request that the authors consider using Repeatability Coefficient (RC) rather than intraclass correlation (ICC) for evaluating test-retest reliability of the same response methods (see Vaz 2013 The Case for Using the Repeatability Coefficient When Calculating Test–Retest Reliability. PLOS ONE). In any case, I would like to see some presentation of data about change-on-retest (how much did individual’s scores change on average? What were the largest changes? Were the changes normally distributed?)

3. Should the authors choose to continue to use ICC to summarize test-retest reliability, I would like to see a justification that includes an explanation of why they have chosen a two-way random model with the test stimuli do not vary among subjects (a one-way random model would be indicated, I think, although this is not at all my area of expertise). I think it is fine to use ICC to compare agreement between the different response paradigms.

Minor issues:

4. The authors should use the term “systematic bias” more judiciously. They are seeking to reduce overall measurement error by minimizing both systematic and random measurement error. As an example, increasing test length serves to reduce random error not systematic bias. Limiting test length avoids the systematic bias on total percent correct scores that would be driven by fatigue.

5. The abstract says: “the paradigm was optimized to account for all variables known to systematically bias task performance.” Be clear with yourself and your readers that the paradigm seeks to reduce measurement error be controlling the stimulus factors know to affect variability. The authors cannot, and do not claim to, control listener health, fatigue, motivation, etc. even though these are variables are known to systematically bias dichotic listening task performance.

6. I think the authors should place more emphasis on the importance of balanced stimulus presentation between the ears (many diagnostic tests using words or digits as stimuli essentially have a list that goes to the right ear and a list to the left ear, so list effects are not controlled). This is a design feature that is a significant improvement on many existing protocols.

7. Similarly I would like to see the authors place more emphasis in the introduction on the comparison of verbal vs manual responses – this is novel and a significant contribution to the literature!

8. I suggest moving from the discussion section into the introduction any references that support the use of dichotic CVs for purpose of measuring hemispheric dominance (e.g., Line 286 to intro “past studies suggest good validity of paradigms using CV stimulus material against more direct indicators of hemispheric dominance”)

9. Pls add a brief limitations section. Acknowledge these results may only be generalizable to educated listeners with symmetrical hearing sensitivity and good diotic scores. The reliability must be replicated in clinical populations, younger/older people, etc.

10. Line 102: I am VERY curious about the person who only identified 81% of diotic stimuli – perhaps this is a person with an auditory processing problem? A person for whom we really want to know their dichotic ability? Pls include in discussion?

11. Section starting with Line 117: Can make this section more parallel with Table 1?

12. Consider adding to discussion – current diagnostic tests are generally short (20-4 pairs). The observation of reliability for a single block of trials is likely closer to the current reliability, but may be better because of the balancing, avoiding negative priming, using stimuli that are more likely to fuse perceptually,

13. Consider adding to discussion – results in line 210 – the effects of practice may be important. Both for reliability, but maybe also for learning (people with APD are unlikely to improve much from a single session of practice? Or people with apd start way low but get better with a little practice?)

14. Line 227-230: This is extremely important result! It suggests that the reliability increase is not due simply to the increase in number of items but rather due to the other factors that were considered in design. Highlight more! Mention it sooner?

Minor suggestions and copy edits:

15. Line 22: “Despite of its popularity” delete the word ‘of’

16. Line 62: “affect the consistency in which individuals perform a given dichotic listening task” wordy. Consider “affect the consistence of individual’s dichotic listening task performance”

17. Line 64: “studies has” to “studies have”

18. Line 43: clarify that the authors are talking about dichotic speech vs dichotic melody or emotion

19. Line 46: “dichotic compared with alternative paradigms” –what are the alternative paradigms that dichotic are better/simpler than? fMRI? Dichotic speech compared to music or emotional content?

20. Line 51-53: If you’re comparing to other paradigms (free-recall in which both are reported or directed attention) do you need to more explicitly acknowledge them?

21. Line 68: “paradigms instructing participants to selectively addend to only one ear at a a time are more difficult” Explain why this is important. There could be situations in which a more difficult test is desirable (e.g., to avoid ceiling effects). (I think you want a test that is driven by perception and difficult tasks are more prone to influence by cognitive factors – make it explicit.)

22. Line 71: Makes me think of tests by Wilson and Moncrieff (random dichotic digits test) and Cameron and Dillon (dichotic digits difference test) that use different approaches to measure the effects of working memory – wonder if that should be addressed in introduction?

23. Line 72: Suggest new paragraph starting with “Ignoring these design variables”

24. Line 73: suggest changing “systematic error variance to the obtained measures” to “measurement error”

25. Line 77: How do we know stimulus driven estimate is about language dominance? Pls explain.

26. Line 82: “All known moderator variables” dial it back – stimulus characteristics or test design variables

27. Line 90: FIRST COMPARISON OF MANUAL VERSUS VERBAL RESPONSE METHOD – THIS IS IMPORTANT!

28. Line 96: Pls start with inclusion/exclusion criteria (Uni Oslo and Bergen, fluent Norwegian, right handed, no known neuor/psych abnormalities, some minimum threshold of hearing sensitivity, symmetrical hearing sensitivity, able to identify stimulus materials diotically… did I get them all?) Then tell us about who presented (52 volunteers) and who was included in the final sample (50), and what criteria the 2 excluded volunteers did not meet. (Or, just leave them out of the conversation all together and just tell us what was true of all 50 participants).

29. Line 106: Pls report absolute hearing sensitivities (did the sample include only those with normal hearing or also those with hearing loss?)

30. Line 141: Was this the number row of a Qwerty keyboard? Were participants able to clearly see the response keys? What happened if they pressed the wrong key (e.g., a “q”)?

31. Lin 146: Blinding the rater was a good idea. Was it possible for the rater to get “off”? (That’s happened to me when scoring CVs on a paper sheet.) Was there any way for the rater to self-correct?

32. Line 151: The laterality index is the only score analyzed, so saying “for further analysis” seems odd. As mentioned before, I think the authors should provide ear specific scores as well.

33. Line 158: “order was balanced between individuals” Does this refer to the response method order? How was this decided? Randomized? Alternating? Listener preference? All manual responses first for the first 24 subjects?

34. Line 162: delete “headphone models” from this list of exact equipment unless Sennheiser HD280 reported above is not accurate.

35. Line 168: In stimuli and procedure the authors failed to indicate the presentation level of the stimuli and whether the stimuli were presented in pseudo-random order or from a pre-ordered list or recording.

36. Line 172: two way mixed models – why not a one way model if the stimuli are the same each time

37. Line 247: “acceptable test length” – acceptable is a subjective assessment. Some listeners may not be able to tolerate 120 items, some clinicians may not be willing to spend 12 min to obtain a laterality estimate (that’s how long it takes me to do a comprehensive audiogram with air, bone, srt, and speech on a compliant adult with symmetric sensorineural hearing loss and normal word recognition – I’m unlikely to add a 12 minute protocol to my routine clinical practice)

Other Comments:

It is outside the scope of this paper, but I am curious if the authors have the data on individual responses to each item – just like there is a “short list” to estimate word recognition ability (stimuli organized by difficulty, if listeners get the first 10 items correct, it is very unlikely they have abnormally low ability, the test can be stopped) I can easily imagine a “short list” dichotic screening (using stimulus dominance information) that is built from the authors’ existing stimuli.

Reviewer #2: This manuscript is interesting and well written.

I ask only for some minor revisions:

Line 20 test or tests? please evaluate

Line 86 cognitive functions

Line 88 perhaps: the total paradigm? or better the entire paradigm

Line 89 allows to evaluate

Line 106 smaller than 10 dB

Lines 102,107 please delete However

Lines 103,108 please delete Furthermore

Line 151 perhaps: analyses

Lines 161, 163 at > in

Results (Lines 189-192): please comment on the relatively high differences in the mean lateralities found. It seems that DL has still some measurements problems.

Line 225: the same lenght.

I strongly agree with the last sentence of the manuscript (lines 285-287) and it would be nice if Authors would comment somewhat further the issue (e.g. concerning the potential use of DL in surgical settings to assess language laterality)

6. PLOS authors have the option to publish the peer review history of their article (what does this mean?). If published, this will include your full peer review and any attached files.

Reviewer #1: Yes: Rocky Kairn Kelley, PhD MS/CCC-A

Reviewer #2: Yes: Alfredo Brancucci

---

## [Author Response · Author response to Decision Letter 0]

28 Apr 2020

Response to reviewer comments, ms PONE-D-20-06948

We would like to thank the reviewers for thoughtful and very encouraging comments, as well as the editor for allow us to revise the manuscript. Please find a point-by-point response below. All changes to the manuscript are marked “red”.

Reviewer #1

Point 1. The authors should include information about listeners’ absolute scores (right ear and left ear) in addition to laterality index. Because LI uses total correct responses (right plus left) as the denominator, for reliability purposes the number of test items is effectively reduced to whatever number the listener got correct [LI can then be modeled as a binomial]. Knowing the total correct scores are important for readers to be able to draw inferences about generalizability. (i.e., the laterality estimate may be reliable among people with high total scores but have much worse reliability among people with more errors, such as a clinical population.)

Response: We have now calculated reliability separately for left and right ear correct responses and corresponding means, sd, and ICCs are provided in Table 2. 

Point 2. I request that the authors consider using Repeatability Coefficient (RC) rather than intraclass correlation (ICC) for evaluating test-retest reliability of the same response methods (see Vaz 2013 The Case for Using the Repeatability Coefficient When Calculating Test–Retest Reliability. PLOS ONE). In any case, I would like to see some presentation of data about change-on-retest (how much did individual’s scores change on average? What were the largest changes? Were the changes normally distributed?)

Response: We were not familiar with the RC, but after consulting the reference paper, we feel that using the “absolute agreement” within the ICC framework accounts as well for both change in mean LI and covariation. However, as suggested, the average change between test 1 and 2 (retest) are now provided in the beginning of the result section. As you can see in the graph below [see uploaded docx version], the distribution in both cases is fairly normal. We are not sure, however, if adding these graphs to the manuscript is really necessary, especially since the scatterplots in Fig 1 nicely (in our opinion) illustrates the modest variability between test time points (i.e. change). 

Point 3. Should the authors choose to continue to use ICC to summarize test-retest reliability, I would like to see a justification that includes an explanation of why they have chosen a two-way random model with the test stimuli do not vary among subjects (a one-way random model would be indicated, I think, although this is not at all my area of expertise). I think it is fine to use ICC to compare agreement between the different response paradigms.

Response: In fact we used a two-way mixed model (not two-way random model), but we have now specified this choice as follows (see Statistical Analysis section): “That is, a two-way model was chosen as test-retest effects were expected (e.g., due to familiarization with the test procedure) so that the second measure might differ systematically from the first. The observations/participants were randomly assigned while the measurements was determined by the research question (fixed), constituting a mixed design.”

Point 4. The authors should use the term “systematic bias” more judiciously. They are seeking to reduce overall measurement error by minimizing both systematic and random measurement error. As an example, increasing test length serves to reduce random error not systematic bias. Limiting test length avoids the systematic bias on total percent correct scores that would be driven by fatigue.

Response: Yes, arguably, we were not stringent in our description. Thus, we carefully reworded the text accordingly.

Point 5. The abstract says: “the paradigm was optimized to account for all variables known to systematically bias task performance.” Be clear with yourself and your readers that the paradigm seeks to reduce measurement error be controlling the stimulus factors know to affect variability. The authors cannot, and do not claim to, control listener health, fatigue, motivation, etc. even though these are variables are known to systematically bias dichotic listening task performance.

Response: Again, we carefully reworded the text accordingly. 

Point 6. I think the authors should place more emphasis on the importance of balanced stimulus presentation between the ears (many diagnostic tests using words or digits as stimuli essentially have a list that goes to the right ear and a list to the left ear, so list effects are not controlled). This is a design feature that is a significant improvement on many existing protocols.

Response: Yes, we agree, this is an important design feature, maybe the most important. We are, however, hesitant in discussing this aspect in more detail here as this would simply reiterate the arguments thoroughly discussed in the referenced review paper (Westerhausen, 2019, Laterality). This review paper discusses this and many other general principles of how to design fair paradigm and is available as open access, i.e. readily available for everyone (https://doi.org/10.1080/1357650X.2019.1598426). In the present manuscript, we thus would rather prefer to focus on the reliability check of the paradigm which was designed accordingly. Table 1 was included to provide an overview of the main criteria we employed designing the paradigm, and the arguments are partially repeated in the method section.. 

Point 7. Similarly I would like to see the authors place more emphasis in the introduction on the comparison of verbal vs manual responses – this is novel and a significant contribution to the literature!

Response: Following the reviewers suggestion, we introduced a new paragraph in the introduction discussing the lack of such comparative studies. 

Point 8. I suggest moving from the discussion section into the introduction any references that support the use of dichotic CVs for purpose of measuring hemispheric dominance (e.g., Line 286 to intro “past studies suggest good validity of paradigms using CV stimulus material against more direct indicators of hemispheric dominance”)

Response: As in response to point 6, we feel Table 1 covers this point sufficiently well considering the recent review paper (Westerhausen, 2019, Laterality) we are referencing here. 

Point 9. Pls add a brief limitations section. Acknowledge these results may only be generalizable to educated listeners with symmetrical hearing sensitivity and good diotic scores. The reliability must be replicated in clinical populations, younger/older people, etc.

10. Line 102: I am VERY curious about the person who only identified 81% of diotic stimuli – perhaps this is a person with an auditory processing problem? A person for whom we really want to know their dichotic ability? Pls include in discussion?

Response: We added the limitation section as requested. However, we are not able to follow up this excluded individual, and any such speculation is certainly beyond what the data allows. The overall laterality measures of this individual were in a normal range (23-64%) and hearing acuity was also unremarkable. Thus, our first interpretation would be that this individual’s diotic results occurred by chance rather than due some underlying pathology. We found the exclusion necessary to make sure the sample is somewhat homogenous. However, including this individual also does not change the present rICC estimates. But we wholeheartedly agree, a reliability analysis in clinical sample should be done before applying it in diagnostics. From a basic research perspective, however, we feel the paradigm is a promising alternative to older established paradigms. 

Point 11. Section starting with Line 117: Can make this section more parallel with Table 1?

Response: We have adjusted Table 1 by numbering and reordering the lines to parallel the order of the design features. We also introduced direct references to the table to make the relationship clearer.

Point 12. Consider adding to discussion – current diagnostic tests are generally short (20-4 pairs). The observation of reliability for a single block of trials is likely closer to the current reliability, but may be better because of the balancing, avoiding negative priming, using stimuli that are more likely to fuse perceptually

Response: We feel any such discussion should be based on a direct comparison in a clinical group. Also, the present paper primarily aims to design a paradigm to assess hemispheric dominance for speech/language processing (as outlined in the introduction and see title). Thus, we would prefer not to delve into this. 

Point 13. Consider adding to discussion – results in line 210 – the effects of practice may be important. Both for reliability, but maybe also for learning (people with APD are unlikely to improve much from a single session of practice? Or people with apd start way low but get better with a little practice?)

Response: Great point! However, like the previous point, without any evidence we would prefer to not take up this discussion. 

Point 14. Line 227-230: This is extremely important result! It suggests that the reliability increase is not due simply to the increase in number of items but rather due to the other factors that were considered in design. Highlight more! Mention it sooner?

Response: We extended the paragraph to highlight this more.

Minor suggestions and copy edits:

Point 15. Line 22: “Despite of its popularity” delete the word ‘of’

Point 16. Line 62: “affect the consistency in which individuals perform a given dichotic listening task” wordy. Consider “affect the consistence of individual’s dichotic listening task performance”

Response: Both corrected as suggested!

Point 17. Line 64: “studies has” to “studies have”

Response: As “a plethora” is singular, we would prefer to keep the following verb in singular too. But interpretations seem acceptable: https://www.merriam-webster.com/words-at-play/plethora-singular-or-plural

Point 18. Line 43: clarify that the authors are talking about dichotic speech vs dichotic melody or emotion 

Point 19. Line 46: “dichotic compared with alternative paradigms” –what are the alternative paradigms that dichotic are better/simpler than? fMRI? Dichotic speech compared to music or emotional content?

Response: Specified a suggested. 

Point 20. Line 51-53: If you’re comparing to other paradigms (free-recall in which both are reported or directed attention) do you need to more explicitly acknowledge them?

Response: We feel the paragraph makes sense as it is right now. But we are obviously explicitly referring to these more demanding paradigms in the next paragraph (now line 67 onwards). 

Point 21. Line 68: “paradigms instructing participants to selectively addend to only one ear at a a time are more difficult” Explain why this is important. There could be situations in which a more difficult test is desirable (e.g., to avoid ceiling effects). (I think you want a test that is driven by perception and difficult tasks are more prone to influence by cognitive factors – make it explicit.)

Response: Absolutely! However, in the mentioned section we list examples for design features and their consequences for the paradigm. And we later specify that the primary aim is to assess hemispheric dominance for language (we now reworked the section to makes this clearer). And here, for sure, any “cognitive component” should be minimised (see next paragraph, lines 74 onwards). We now reworked abstract and introduction to more clearly emphasise that the primary purpose of the new paradigm is to assess hemispheric dominance.

Point 22. Line 71: Makes me think of tests by Wilson and Moncrieff (random dichotic digits test) and Cameron and Dillon (dichotic digits difference test) that use different approaches to measure the effects of working memory – wonder if that should be addressed in introduction?

Response: Rather feel that would distract from the main purpose. But in general it would be a great idea to have a systematic look at the paradigms used in APD and other speech impediment diagnostics under consideration of the background (Westerhausen, 2019, Laterality) paper. If the reviewer would be interested we attempt doing a joined critical review on what is done. 

Point 23. Line 72: Suggest new paragraph starting with “Ignoring these design variables”

Point 24. Line 73: suggest changing “systematic error variance to the obtained measures” to “measurement error”

Response: Both done as suggested.

Point 25. Line 77: How do we know stimulus driven estimate is about language dominance? Pls explain.

Response: We have include a section into the introduction elaborating the assumed theoretical framework (line 74 onwards), i.e. our understanding of dichotic listening. A more detailed outline can be found in another recent reference (Westerhausen & Kompus, 2018, Scan J Psychol; https://doi.org/10.1111/sjop.12408). 

Point 26. Line 82: “All known moderator variables” dial it back – stimulus characteristics or test design variables

Response: “Dialed down”

Point 27. Line 90: FIRST COMPARISON OF MANUAL VERSUS VERBAL RESPONSE METHOD – THIS IS IMPORTANT!

Response: Thanks for sharing our enthusiasm for this work. As indicated in response to point 7, a new paragraph was added to the introduction appreciating the importance of the response format comparison. 

Point 28. Line 96: Pls start with inclusion/exclusion criteria (Uni Oslo and Bergen, fluent Norwegian, right handed, no known neuor/psych abnormalities, some minimum threshold of hearing sensitivity, symmetrical hearing sensitivity, able to identify stimulus materials diotically… did I get them all?) Then tell us about who presented (52 volunteers) and who was included in the final sample (50), and what criteria the 2 excluded volunteers did not meet. (Or, just leave them out of the conversation all together and just tell us what was true of all 50 participants). 

Point 29. Line 106: Pls report absolute hearing sensitivities (did the sample include only those with normal hearing or also those with hearing loss?)

Response: we restructured the section accordingly and absolute hearing info was provided. We prefer to report all participants recruited and tested for matter of completeness. 

point 30. Line 141: Was this the number row of a Qwerty keyboard? Were participants able to clearly see the response keys? What happened if they pressed the wrong key (e.g., a “q”)?

Response: As written in text it was the number pad (not the row) and there was no possibility to correct the response (also to avoid long thinking about the response, which might alter things) as now indicated in text. The buttons were clearly marked.

Point 31. Lin 146: Blinding the rater was a good idea. Was it possible for the rater to get “off”? (That’s happened to me when scoring CVs on a paper sheet.) Was there any way for the rater to self-correct?

Response: No possibility to correct either, as the same response set up was used as for the manual response. Possible “rater errors” we considered as “noise” in the system but considering the high number of correct answers (see Table 2), cannot have been frequent. 

Point 32. Line 151: The laterality index is the only score analyzed, so saying “for further analysis” seems odd. As mentioned before, I think the authors should provide ear specific scores as well.

Response: we now provide the analysis of the correct left and right ear score

Point 33. Line 158: “order was balanced between individuals” Does this refer to the response method order? How was this decided? Randomized? Alternating? Listener preference? All manual responses first for the first 24 subjects?

Response: Yes, response format; and “odd-even” split to get the same amount of individuals starting with verbal and manual response

Point 34. Line 162: delete “headphone models” from this list of exact equipment unless Sennheiser HD280 reported above is not accurate.

Response: Yes, same model at each site. Thanks for spotting!

Point 35. Line 168: In stimuli and procedure the authors failed to indicate the presentation level of the stimuli and whether the stimuli were presented in pseudo-random order or from a pre-ordered list or recording.

Response: information now provided (line 158)

Point 36. Line 172: two way mixed models – why not a one way model if the stimuli are the same each time

Response: Please refer to point 3

point 37. Line 247: “acceptable test length” – acceptable is a subjective assessment. Some listeners may not be able to tolerate 120 items, some clinicians may not be willing to spend 12 min to obtain a laterality estimate (that’s how long it takes me to do a comprehensive audiogram with air, bone, srt, and speech on a compliant adult with symmetric sensorineural hearing loss and normal word recognition – I’m unlikely to add a 12 minute protocol to my routine clinical practice)

Response: We were more thinking of experiments. Thanks for pointing this out, we specified this comment now to read “at least in experimental settings”. But for the assessment of LIs, our data suggests that 80 trials, so about 8 mins, seem to be the minimum (in healthy indivdiuals)

Other Comments:

It is outside the scope of this paper, but I am curious if the authors have the data on individual responses to each item – just like there is a “short list” to estimate word recognition ability (stimuli organized by difficulty, if listeners get the first 10 items correct, it is very unlikely they have abnormally low ability, the test can be stopped) I can easily imagine a “short list” dichotic screening (using stimulus dominance information) that is built from the authors’ existing stimuli.

Response: We are happy to share the excel files of the trial by trial response if you would like to explore the data deeper. Although the experiment only consists of 12 different stimulus pairs, and if one wants to avoid negative priming one might be able to reduce it to 8 pairs. One pair for each voicing combination and presented in the left-right and right-left order. 

Reviewer #2 

Line 20 test or tests? please evaluate

Response: “Tests”, thanks for pointing out the inconsistency

Line 86 cognitive functions

Response: Sentence removed in revision

Line 88 perhaps: the total paradigm? or better the entire paradigm

Line 89 allows to evaluate 

Line 106 smaller than 10 dB

Lines 102,107 please delete However

Lines 103,108 please delete Furthermore

Line 151 perhaps: analyses

Lines 161, 163 at > in

Response: corrected as suggested

Results (Lines 189-192): please comment on the relatively high differences in the mean lateralities found. It seems that DL has still some measurements problems.

Response: This translates to the small main effect of Repetition we find in the ANOVA, which we accordingly discuss. 

Line 225: the same lenght.

Response: corrected, thanks again

I strongly agree with the last sentence of the manuscript (lines 285-287) and it would be nice if Authors would comment somewhat further the issue issue (e.g. concerning the potential use of DL in surgical settings to assess language laterality) 

Response: Thanks! If one should use the paradigm for planning surgery would be more question of validity than reliability, which we now discuss in the “limitations” paragraph. However, review one finds the paradigm too long for being of use in audiological diagnostics.

---

## [Decision Letter · Decision Letter 1]

1 Jun 2020

An optimal dichotic-listening paradigm for the assessment of hemispheric dominance for speech processing

PONE-D-20-06948R1

Dear Dr. Westerhausen,

We are pleased to inform you that your manuscript has been judged scientifically suitable for publication and will be formally accepted for publication once it complies with all outstanding technical requirements.

With kind regards,

Simone Sulpizio

Academic Editor

PLOS ONE

Additional Editor Comments (optional):

Reviewers' comments:

Reviewer's Responses to Questions

**Comments to the Author**

1. If the authors have adequately addressed your comments raised in a previous round of review and you feel that this manuscript is now acceptable for publication, you may indicate that here to bypass the “Comments to the Author” section, enter your conflict of interest statement in the “Confidential to Editor” section, and submit your "Accept" recommendation.

Reviewer #1: All comments have been addressed

2. Is the manuscript technically sound, and do the data support the conclusions?

Reviewer #1: Yes

3. Has the statistical analysis been performed appropriately and rigorously? 

Reviewer #1: Yes

4. Have the authors made all data underlying the findings in their manuscript fully available?

Reviewer #1: Yes

5. Is the manuscript presented in an intelligible fashion and written in standard English?

Reviewer #1: Yes

6. Review Comments to the Author

Reviewer #1: This is an excellent manuscript and a valuable new test paradigm for improved measurement of dichotic listening. The authors have met or exceeded all standards required for publication.

7. PLOS authors have the option to publish the peer review history of their article (what does this mean?). If published, this will include your full peer review and any attached files.

Reviewer #1: No

---

## [Editor Report · Acceptance letter]

3 Jun 2020

PONE-D-20-06948R1 

An optimal dichotic-listening paradigm for the assessment of hemispheric dominance for speech processing 

Dear Dr. Westerhausen:

I'm pleased to inform you that your manuscript has been deemed suitable for publication in PLOS ONE. Congratulations! Your manuscript is now with our production department. 

Kind regards, 

on behalf of

Dr. Simone Sulpizio 

Academic Editor

PLOS ONE